# ONLINE LEARNING FOR OBSTACLE AVOIDANCE

## ABSTRACT

We approach the fundamental problem of obstacle avoidance for robotic systems via the lens of online learning. In contrast to prior work that either assumes worst-case realization of uncertainty in the environment or a given stochastic model of uncertainty, we propose a method that is efficient to implement and provably grants *instance-optimality* to perturbations of trajectories generated from an open-loop planner in the sense of minimizing worst-case regret. The resulting policy thus adapts online to realizations of uncertainty and provably compares well with the best obstacle avoidance policy *in hindsight* from a rich class of policies. The method is validated in simulation on a dynamical system environment and compared to baseline open-loop planning and robust Hamilton-Jacobi reachability techniques.

## 1 INTRODUCTION

The problem of obstacle avoidance in motion planning is a fundamental and challenging task at the core of robotics and robot safety. Successfully solving the problem requires dealing with environments that are inherently uncertain and noisy: a robot must take into account uncertainty in its own dynamics, e.g., due to external disturbances or unmodeled effects, and the dynamics of other agents in the environment, e.g., humans or other robots. Approaches for tackling the obstacle avoidance problem in robotics typically fall under two categories: (i) methods that attempt to construct stochastic models of uncertainty in the dynamics of the robot and other agents and use the resulting probabilistic models for planning, and (ii) methods that construct plans that take into account worst-case behavior. In Sec. 2 we give a more detailed overview of both classes of approaches.

In this paper, we are motivated by Vapnik's principle: *"when solving a given problem, try to avoid solving an even harder problem as an intermediate step"*. Constructing accurate models of external disturbances and the dynamics of humans or other agents is perhaps more complicated than the task of obstacle avoidance in motion planning. The uncertainty in these dynamics rarely conform to the assumptions made by the two classes of approaches highlighted above: external disturbances and human motion are typically neither fully stochastic nor fully adversarial. This motivates the need for *online learning* methods for *adaptive non-stochastic* control to avoid obstacles as they are perceived by the robot's sensors.

**Statement of Contributions.** In this work, we pose the problem of obstacle avoidance in a *regret minimization* framework and build on techniques from non-stochastic control. Our primary contribution is a gradient-based online learning algorithm for the task of obstacle avoidance, coupled with provable regret bounds that show our obstacle avoidance policy to be comparable to the *best policy in hindsight* from a given class of closed-loop policies. This type of theoretical performance guarantee is nonstandard, and allows us to flexibly adapt to the behavior of the uncertainty in any *instance* of the obstacle avoidance problem without making *a priori* assumptions about whether the uncertainty is stochastic or adversarial. In addition, the resulting method is computationally efficient. The method is applied to experiments with complex (and dynamic) obstacle environments, and demonstrates improved performance in respective instances where open-loop planners and overly-robust methods can struggle.

## 2 RELATED WORK

Our results apply techniques from online learning to the problem of obstacle avoidance in order to provide efficiently computable safety guarantees. Both online learning and safe motion planning

have generated significant interest and theoretical development in the past few years; we set our work within these contexts below.

**Online Learning for Control.** Many canonical control-theoretic results have recently been cast as problems in online learning (Hannan, 1958; Cesa-Bianchi & Lugosi, 2006), providing interesting generalizations to established control results like the linear-quadratic regulator (Cohen et al., 2018; Agarwal et al., 2019a;c) and $H_\infty$ robust control (Ghai et al., 2021). Optimal results in sample complexity (Dean et al., 2020) and control synthesis for unknown linear systems (Hazan et al., 2020) illustrate further generalizations of standard controllers.

Control formulations are often efficiently solvable due to the convexity of the objective; however, "higher-level" decision-making tasks like motion planning and obstacle avoidance have non-convex objective functions (e.g., maximizing the distance of an ego vehicle to its nearest obstacle). Fortunately, some non-convex problem formulations actually admit "hidden convexity," meaning that they can be reformulated (via nonlinear transformations, relaxations, or conversions to an efficient dual formulation — see Xia (2020) for a survey) into equivalent optimization problems that *are convex*. This allows for efficient solutions (e.g., Ben-Tal & Teboulle (1996); Konno & Kuno (1995); Voss et al. (2016)) to problems that nominally would be hard to solve (e.g., Matsui (1996)). Our work gives such a reformulation for an obstacle avoidance task, which differs from but builds upon the existing online learning and control literature.

**Motion Planning.** Effective motion planning is a central challenge within robotics that has both generated consistent interest for many years and has spurred many recent developments. Inspired by the classical search algorithms of Dijkstra (1959); Hart et al. (1968), many sampling-based, probabilistic techniques (Kavraki et al., 1996; LaValle & Kuffner, 2001; Karaman & Frazzoli, 2011; Kobilarov, 2012) have come to the fore due to several key characteristics: (1) provable probabilistic completeness, (2) extension to continuous state, action, and configuration spaces, and (3) direct integration with differential constraints on robot kinematics and dynamics. The case of obstacle avoidance in multi-agent settings has also been considered in the literature, although many existing practical algorithms (van den Berg et al., 2011; Zhou et al., 2017) are difficult to analyze theoretically. Most importantly, however, none of the above methods directly model and account for the effects of disturbances, sensor noise, or other uncertainty within the system.

**Safe and Robust Motion Planning.** Construction of robust planning algorithms forms the natural development from the above limitation. Existing robust planning techniques generally fall into one of two categories: (i) methods that make assumptions on the distribution of uncertainty, or (ii) methods that assume worst-case disturbances. One of the earliest methods utilizes chance constraints to bound the probability of collision under stochastic uncertainty (Charnes & Cooper, 1959); this framework is extended to various sources of stochastic uncertainty in dynamical systems in Blackmore et al. (2006); Du Toit & Burdick (2011). To model stochastic noise in a robot's sensory observations, extensions of planning algorithms to partially observable Markov decision processes (POMDPs) and to planning in the robot belief space have been developed (Kaelbling et al., 1998; Prentice & Roy, 2009; Platt et al., 2012). However, these methods do not account for a notion of worst-case behavior yet must often still rely on moderately strong distributional assumptions (e.g. Gaussian noise) in practice.

To account for worst-case disturbances, Hamilton-Jacobi (HJ) reachability (Mitchell & Tomlin, 2003; Bansal et al., 2017) and robust trajectory library methods (Singh et al., 2017; Majumdar & Tedrake, 2017) have been developed to provide non-stochastic (adversarial) certificates of safety for path planning problems. Given only an assumption of bounded disturbances, these techniques construct corresponding representations (or outer approximations) to the safe and unsafe regions of the state space conditional on the robot dynamics, obstacle placement, and disturbance size. HJ methods, which generally suffer from the curse of dimensionality (Darbon & Osher, 2016) (although with the potential for speed-ups in certain settings (Herbert et al., 2017)), use the formalism of the differential game (Isaacs, 1965) to provide a "global" notion of safe and unsafe sets (Mitchell et al., 2005). By comparison, robust trajectory libraries, which are usually computed using convex programs (Parrilo, 2003) provide safety guarantees in the form of robust "tubes" or "funnels" (Burridge et al., 1999; Majumdar & Tedrake, 2017) that encompass only the nominal trajectory (or hypothesized trajectories) within the space.

The key distinction of our work from these methods arises from the desire for "instance-optimality" — namely, the development of an algorithm through the use of online learning and regret minimization

that yields a controller that adapts to the nature of the disturbances observed. In particular, this allows the method to recover effective performance in *both* stochastic and non-stochastic regimes; we do not sacrifice good transient behavior in "benign" environments in order to provide guarantees about robust performance in more adversarial cases, and vice versa.

## 3 PROBLEM FORMULATION

Consider a discrete-time dynamical system with state $\bar{\mathbf{x}}$ and control input $\bar{\mathbf{u}}$. A planning oracle $\mathcal{O}_\mathcal{T}(\bar{\mathbf{x}}_0)$ takes in an initial state and generates a nominal state trajectory with associated control inputs $\mathcal{T} = \{\bar{\mathbf{x}}_t^0, \bar{\mathbf{u}}_{t-1}^0\}_{t=1}^T$. We intend to design an adversary-robust obstacle avoidance controller that will act to update the trajectory to avoid local obstacles. Intuitively, this will serve as a faster "safety inner loop" for the slower trajectory planning stage $\mathcal{O}_\mathcal{T}$, keeping the agent safe from external features unseen at planning time. The key modeling assumption necessary to proceed is to assume that the dynamics of perturbations about the nominal trajectory are well-modeled by linear discrete-time dynamics[1]. Defining $\mathbf{x}_t = \bar{\mathbf{x}}_t - \bar{\mathbf{x}}_t^0$ and $\mathbf{u}_t = \bar{\mathbf{u}}_t - \bar{\mathbf{u}}_t^0$, we express this modeling assumption as

$$\mathbf{x}_{t+1} = A\mathbf{x}_t + B\mathbf{u}_t + D\mathbf{w}_t, \tag{1}$$

where $\mathbf{w}_t$ is a bounded, unknown, possibly-adversarial disturbance. For many practical cases, the linear dynamics assumption is reasonable; one concrete example is a control-affine system with feedback linearization in the planning controller. Our task is to construct a "residual" controller generating $\mathbf{u}_t$ so as to correct the system in order to avoid obstacles. Intuitively, $\mathcal{O}_\mathcal{T}$ is the optimistic, goal-oriented planner (in practice, the planner could be an off-the-shelf algorithm, see for example the OMPL library (Sucan et al., 2012)), while our controller acts as a robust safety mechanism that becomes active *only when needed, and in a provably effective manner*.

### 3.1 SAFETY CONTROLLER OBJECTIVE

The proxy objective for the controller trying to avoid obstacles is to maximize the distance to the nearest obstacle, subject to regularization of state deviations and control usage. We assume a sensor mechanism that reports all obstacle positions within a given radius of the agent. The optimization problem, for a trajectory of length $T$, safety horizon of length $L \leq T$, and obstacle positions $\mathbf{p}_t^j$ denoting the $j^{th}$ sensed obstacle at time $t$, is

$$\max_{A \in \mathcal{A}} \left\{ C_{obs}(A) = \sum_{t=1}^T \min_{\tau \in [L]} \min_j \|\mathbf{x}_{t+\tau}^A - \mathbf{p}_t^j\|_2^2 - \|\mathbf{x}_t^A\|_Q^2 - \|\mathbf{u}_t^A\|_R^2 \right\}. \tag{2}$$

Here, $\mathcal{A}$ is the set of online algorithms that choose actions for the controller to take, and $\mathbf{x}_t^A$ denotes the state trajectory reached given actions $\mathbf{u}_t^A \sim A \in \mathcal{A}$. The last two terms represent quadratic state and action costs penalizing large deviations from the nominal trajectory and large actions; these costs[2] are very common objectives in the control literature, but serve in this instance instead to regularize the solution to the obstacle avoidance task. Further, in order to be tractably analyzed, the collision avoidance objective is relaxed to a quadratic penalty term; nonetheless, even this relaxation is nonconvex due to the selection over time and obstacle indices of the minimal-distance obstacle in the first term of $C_{obs}$. As the optimal policy may be hard to find in practice, we instead relax the problem into one of online learning and regret minimization (Agarwal et al., 2019a). This allows us to define the safety metric with respect to the best achievable performance in hindsight with respect to a comparator class (which we describe in the following section). For a sufficiently powerful and convex comparator class, we can achieve guarantees on the safety of the resulting controller.

### 3.2 REGRET FRAMEWORK FOR OBSTACLE AVOIDANCE

Leveraging the expressiveness of linear dynamic controllers (LDCs) (Ghai et al., 2021; Agarwal et al., 2019a), we use the disturbance-action controller parameterization to construct a low-regret

---

[1]The results here are presented for linear time-invariant (LTI) systems. We believe that the results extend to linear time-varying (LTV) systems under reasonable but slightly more technical assumptions; however, this formal analysis is left as future work.

[2]Here, we omit the fully general LQR costs (time-varying $Q_t$ and $R_t$) for simplicity of presentation; however, the results we show will *also* hold for this more general setting.

framework. The action is the sum of a stabilized nominal action and a residual obstacle-avoiding term, as shown in Eqn. 3:

$$\mathbf{u}_t = K\mathbf{x}_t + \sum_{i=1}^{H} M^{[i]}\mathbf{w}_{t-i}. \tag{3}$$

It will be useful to define the intermediate quantities $\tilde{\mathbf{u}}_t = \mathbf{u}_t - K\mathbf{x}_t$ and $\tilde{A} = A + BK$. Then the system dynamics are equivalently expressed as

$$\mathbf{x}_{t+1} = \tilde{A}\mathbf{x}_t + B\tilde{\mathbf{u}}_t + D\mathbf{w}_t. \tag{4}$$

The comparator class $\Pi$ will be the class of LDCs parameterized by $M$; this class has provably expressive approximation characteristics (Agarwal et al., 2019a). The regret is defined using quantities from Eqn. 2 as

$$\text{Reg}_T(A) = \sup_{w_1,\ldots,w_T} \left\{ \max_{M\in\Pi} C_{obs}(M) - C_{obs}(A) \right\}. \tag{5}$$

Eqn. 5 implies that the adaptive sequence $M_t$ selected by low-regret online algorithm $A$ will compete on average with the best fixed policy $M^*$ in hindsight. Note that this is an instance-optimal, finite time guarantee. Specifically, the optimum $M^*$ may vary with the choice of disturbances, the selection of time horizon T, or the particular obstacle array and planner. The guarantee ensures that the sequence of $M_t$ chosen by algorithm $A$ competes with $M^*$ *for all* choices of these parameters.

### 3.3 TRUST REGION OPTIMIZATION

A trust region optimization problem (Sorensen, 1980) is a nominally non-convex optimization problem that admits hidden convexity. Formulating a problem as a trust region instance (see Def. 1) allows for a convex relaxation that can be solved efficiently (Ben-Tal & Teboulle, 1996; Hazan & Koren, 2016).

**Definition 1** (Trust Region Instance). *A trust region instance is defined by a tuple* $(P, \mathbf{p}, D)$ *with* $P \in \mathbb{R}^{d\times d}$, $\mathbf{p} \in \mathbb{R}^d$, *and* $D > 0$ *as the mathematical optimization problem*

$$\max_{\|\mathbf{z}\|\leq D} \left\{ \mathbf{z}^T P \mathbf{z} + \mathbf{p}^T \mathbf{z} \right\}. \tag{6}$$

Throughout the remainder of this paper, we will use "trust region instance" to refer interchangeably to instances of Def. 1 and the implicit, equivalent convex relaxation.

## 4 METHODOLOGY AND ALGORITHM

We begin with a brief outline of the regret minimization method, from which we outline the resulting algorithm and describe its key properties.

### 4.1 REGRET MINIMIZATION METHODOLOGY

Many existing results in online learning and control formulate guarantees by posing the specific problem as a (possibly complex) online convex optimization (Hazan, 2021) instance, which can be solved in an efficient manner (Boyd & Vandenberghe, 2004). This method alone will not suffice for our setting because of the non-convexity of the objective. However, optimal regret bounds for online *non-convex* optimization have been shown via a "Follow the Perturbed Leader" (FPL) method for problems with an optimization oracle of known performance (Agarwal et al., 2019b; Suggala & Netrapalli, 2020). This was extended to the setting with memory in Ghai et al. (2021). The key challenge here is to reduce our non-convex setting to one in which a good oracle is feasible, and then to follow through with the analysis to combine the results of Suggala & Netrapalli (2020); Ghai et al. (2021) with the guarantees for online convex optimization.

### 4.2 NON-CONVEX MEMORY FPL FOR OBSTACLE AVOIDANCE

Adapting the general online non-convex FPL framework, we formulate the obstacle avoidance safety controller as an online FPL algorithm, shown in Alg. 1. The procedure is relatively direct. The agent

plays $M_t$, which induces $\tilde{\mathbf{u}}_t$ via Eqn. 3. The state dynamics are propagated to reveal $\mathbf{w}_t$ and observe $\mathbf{x}_{t+1}$, and then a sensor reading is taken to observe the new local obstacle positions. The reward (Eqn. 7) is revealed *by the environment*, and the agent incurs the loss of action $M_t$. Finally, the agent updates $M_{t+1}$ using the new information by solving the optimization problem in Eqn. 8. Eqn. 8 is the Follow-the-Perturbed-Leader component of the algorithm, where the inner product $(M \bullet P_0)$ serves as the regularizing perturbation. Importantly, the history length $H$ determines how large $M$ is (see Eqn. 3). We will show that $H$ can be reduced to logarithmic dependence on $T$, so the random play at the outset in Alg. 1 does not form a significant portion of the game.

---

**Algorithm 1** Non-convex Memory FPL for Obstacle Avoidance

---

**Input**: (partially observed) $\{\mathbf{p}_t^j\}_{j=1}^K$, planning horizon $L$, history length $H$

**Input**: Full horizon $T$, algorithm parameters $\{\eta, \epsilon, \lambda\}$, initial state $\mathbf{x}_0$

**Input**: Open-loop plan: $\bar{\mathbf{u}}_t^o$ for $t = 1, ..., T$

**Initialize**: Closed-loop correction $M_0^{[1:H]}$, fixed perturbation $P_0 \sim \text{Exp}(\eta)^{d_u \times H d_w}$

**Initialize**: Play randomly for $t = \{0, ..., H-1\}$, observe rewards, states, noises, and obstacles

**for** t=H...T-1 **do**

   Play $M_t^{[1:H]}$ and observe state $\mathbf{x}_{t+1}$, noise $\mathbf{w}_t$, and sensed obstacles $\{\mathbf{p}_{t+1}^j\}_{j \in [k]_{t+1}}$.

   Observe the reward function

$$\ell_{t+1}(M_t^{[1:H]}) = \min_{j \in [k]} \left\{ \|\mathbf{x}_{t+1} - \mathbf{p}_{t+1}^j\|_2^2 \right\} - \|\mathbf{x}_{t+1}\|_Q^2 - \|\tilde{\mathbf{u}}_t\|_R^2. \tag{7}$$

   Solve for $M_{t+1}^{[1:H]}$ as the solution to

$$[M^*]_{t+1}^{[1:H]} = \underset{\|M\| \leq D_M}{\arg\max} \left\{ \sum_{\tau=1}^{t+1} \ell_\tau(M) + \lambda(M \bullet P_0) \right\}. \tag{8}$$

**end for**

**return** $M_{T+1}$

---

To feasibly engage the obstacle avoidance algorithm, it is necessary to solve for optimal solutions to the objective in Eqn. 8. We show that the optimum can be found as the solution to an iterative "hidden convex"-concave game (Hazan, 2006) in which the players respectively solve a hidden-convex trust region objective and utilize a low-regret algorithm (exponentiated gradient from Kivinen & Warmuth (1997)) for a linear objective on the simplex. We show in Alg. 2 a method to iteratively solve for $M_{t+1}^{[1:H]}$ in Eqn. 8 of Alg. 1.

---

**Algorithm 2** (General) Hidden-Convex Formulation for Objective in Eqn. 8

---

**Input**: Set of vectors $\{\mathbf{a}_j\}_{j=1}^k$, matrix $B$, vectors $\mathbf{b}, \mathbf{b}_0$

**Input**: Iterations $N$, learning rate $\eta$, approx. error $\epsilon$, perturbation $P_0$, diameter $D_M$

**Initialize**: Vector $\mathbf{c}_0 = \frac{1}{k}\mathbb{1}^k$

**for** n=0...N **do**

   (1) Solve for $M_n$

$$M_n = \underset{\|M\| \leq D_M}{\arg\max} \left\{ \sum_{j=1}^k \mathbf{c}_n(j)\|\mathbf{a}_j + BM\mathbf{b}\|_2^2 - \|\mathbf{b}_0 + BM\mathbf{b}\|_Q^2 - \|M\mathbf{b}\|_R^2 + \lambda(M \bullet P_0) \right\} \tag{9}$$

   (2) Update $\mathbf{c}_{n+1}$

$$\mathbf{c}_{n+1} = \underset{\Delta_k}{\Pi} \left[ \mathbf{c}_n e^{-\eta \nabla_{\mathbf{c}} \left( \sum_j \mathbf{c}_n(j)\|\mathbf{a}_j + BM\mathbf{b}\|_2^2 \right)} \right] \tag{10}$$

**end for**

**return** $M_N$

---

## 5 ANALYSIS AND REGRET BOUND

We now present our main theoretical contributions: we demonstrate that Alg. 2 efficiently solves the optimization in Eqn. 8, and that (as a consequence) Alg. 1 attains low (sublinear) regret. We defer details of all proofs to App. A and only present the main ideas here.

### 5.1 CONVEX OBSTACLE AVOIDANCE: ALG. 2 EFFICIENTLY SOLVES EQN. (8)

We first analyze the behavior of Alg. 2 and demonstrate that $M_N$ solves the optimization problem (8). To establish this we need a few assumptions and a key lemma.

**Assumption 2** (Bounded Disturbances). *The disturbances are norm-bounded by a constant $C_w$.*

**Assumption 3** (Stabilizability and Conditioning). *The matrix $\tilde{A} = H\Lambda H^{-1}$, with $\|\Lambda\| \leq 1 - \gamma$. Further, assume that $\|\tilde{A}\|\|\tilde{A}^{-1}\| \leq \kappa$ and that $\|\tilde{A}\|, \|B\|, \|D\| \leq \beta$.*

**Remark 4** (Bounded Behavior). *The state $(\mathbf{x}_t)$ and input $(\tilde{\mathbf{u}}_t)$ are norm-bounded by constants $C_x, C_u$.*

*Proof.* From Assumption 2 and Eqn. 3 we have that $\|\tilde{\mathbf{u}}_t\| \leq HD_M C_w$. Therefore, by Assumption 3, we have that $\|B\tilde{\mathbf{u}}_t + D\mathbf{w}_t\| \leq 2\beta HD_M C_w$. Assume wlog that $\|\mathbf{x}_0\| \leq 2\beta HD_M C_w\gamma^{-1}$. Then it is true that for every $t \geq 0$, $\|\mathbf{x}_t^{\tilde{A}}\|_2 \leq 2\beta HD_M C_w\gamma^{-1}$. Then $C_x, C_u$ can be chosen appropriately. □

**Assumption 5** (No Extraneous Inputs). *The dynamics matrix $B \in \mathbb{R}^{d_x \times d_u}$ has $rank(B) = d_u$.*[3]

**Lemma 6** (Trust Region Instance). *Eqn. (9) is a trust region instance (Defn. 1) that can be solved efficiently.*

Proof of this lemma is deferred to Appendix A.4.

**Theorem 7** (Solving Eqn. (8)). *The optimization in Eqn. (8) is solved efficiently using Alg. 2.*

*Proof.* The proof requires two steps: constructing an instance of Eqn. (9) that solves Eqn. (8), and demonstrating that Alg. 2 converges to this instance and its optimal solution by using Lemma 6. For space considerations, the full construction is deferred to App. A.2 □

### 5.2 TOWARDS A REGRET BOUND: FPL WITH MEMORY

In order to demonstrate the low regret of Alg. 1, we utilize a regret bound that extends results for non-convex FPL to the equivalent setting *with memory* (Anava et al., 2015; Arora et al., 2012). This is important for the control theoretic setting we consider here, where there is inherent memory of previous control inputs expressed in the state of the agent. This extension of non-convex FPL to settings with memory was developed in Ghai et al. (2021); we state their result here.

**Theorem 8** (FPL with Memory Ghai et al. (2021)). *Let $\mathcal{M} \in \mathbb{R}^d$ be a convex decision set with $\|\mathcal{M}\|_\infty \leq D$. Assume that for all $t \in [T]$, the loss functions $\ell_t(M_i, M_j, ..., \cdot)$ are L-Lipschitz with respect to the $\ell_1$-norm in each argument. Then, for any $\eta$, there exists an algorithm (see App. A.3) that achieves expected regret*

$$\max_{M \in \mathcal{M}} \sum_{t=H}^T \ell_t(M, ..., M) - \mathbb{E}\Big[\sum_{t=H}^T \ell_t(M_{t-H+1}, ..., M_t)\Big] \leq L^2 H^3 d^2 D\eta T + \frac{dD}{\eta} + \varepsilon HT.$$

### 5.3 TOWARDS A REGRET BOUND: STATE APPROXIMATION VIA STABILITY AND HISTORY

In order to apply the results from Sec. 5.2 for establishing low regret of Alg. 1, we need to limit the effect of the state history on the regret. This is accomplished via the stability of the closed-loop dynamics. Since we use the disturbance-action parameterization (Agarwal et al., 2019a; Hazan et al.,

---

[3]Assumption 5 holds without loss of generality; if $rank(B) = r_B < d_u$, we simply choose a subset of $B$ with $r_B$ independent columns and fix any other inputs to be uniformly 0.

2020) for the controller in Eqn. (3), it can be shown that the state can be expressed as a function of a limited history of the actions:

$$\mathbf{x}_{t+1}^{\mathcal{A}} = \tilde{A}^{H+1}\mathbf{x}_{t-H}^{\mathcal{A}} + \sum_{j=0}^{2H} \Psi_{t,j}\mathbf{w}_{t-j} \approx \sum_{j=0}^{2H} \Psi_{t,j}\mathbf{w}_{t-j} := \hat{\mathbf{x}}_{t+1}^{\mathcal{A}}, \tag{11}$$

where $\Psi_{t,j}$ is the disturbance-state transfer function (matrix) between disturbance $\mathbf{w}_{t-j}$ and state $\mathbf{x}_{t+1}^{\mathcal{A}}$ (see Eqn. 12). The boundedness of $H$ ensures that the effect of any "bad" action does not linger too long within the system. By the $(1-\gamma)$-stability of $\tilde{A}$ (Assumption 3), for the state to be approximated by Eqn. (11) within an error fraction of $\epsilon$ the history $H$ grows only logarithmically with $1/\epsilon$. Further, note that $\Psi_{t,j}$ is linear in $M_\tau$ for all $\{t, \tau, j\}$:

$$\Psi_{t,j}(M_{t-H}, ..., M_{t-1}) = \tilde{A}^j D\mathbb{1}[j \leq H] + \sum_{i=1}^{H} \tilde{A}^{i-1} M_{t-i}^{[j-i]} \mathbb{1}[j-i \in \{1, 2, ..., H\}]. \tag{12}$$

**Remark 9** (Bounded Cost Error). *For $H = \lceil \gamma^{-1} \log \left( 5\kappa^2 C_x(1 + \beta D_M C_w)T \right) \rceil$, the error in cost is less than $C_x/T$. See App. A.5.2 for the proof.*

### 5.4    REGRET BOUND

We conclude the analysis by stating the main theorem of the paper, which establishes the low regret of the obstacle avoidance algorithm.

**Theorem 10** (Regret Bound for Alg. 1). *Alg. 1, utilizing Alg. 2 as an optimization subroutine, achieves*

$$Reg_T(A) \leq \tilde{\mathcal{O}}(poly(\mathcal{L})\sqrt{T}), \tag{13}$$

*where $\mathcal{L}$ is a measure of the complexity of the optimization instance (Ghai et al., 2021; Agarwal et al., 2019a; Chen & Hazan, 2021).*

*Proof.* Using the boundedness of the disturbances, states, and actions (Assumption 2 and Remark 4), we choose an appropriate $H$ such that the error of approximation of the state $\hat{\mathbf{x}}_{t+1}^{\mathcal{A}}$ is small per Eqn. (11). App. A.5.2 gives $H = \mathcal{O}(\log T)$ such that the error in associated costs is less than $C_x/T$ (see Remark 9).

Now, we analyze the regret as follows:

$$\text{Reg}_T(A) := \max_{M \in \Pi} \sum_{t=H-1}^{T-1} \ell_{t+1}(\mathbf{x}_t^M, \{\mathbf{p}_t^j\}, \mathbf{u}_t(M)) - \sum_{t=H-1}^{T-1} \ell_{t+1}(\mathbf{x}_t^{\mathcal{A}}, \{\mathbf{p}_t^j\}, \mathbf{u}_t(\mathcal{A}))$$

$$\leq \max_{M \in \Pi} \sum_{t=H-1}^{T-1} \left( \ell_{t+1}(M, M, ..., M) + \frac{C_x}{T} \right) - \sum_{t=H-1}^{T-1} \left( \ell_{t+1}(M_{t-H}, ..., M_t) - \frac{C_x}{T} \right)$$

$$= \left[ \max_{M \in \Pi} \sum_{t=H-1}^{T-1} \ell_{t+1}(M, M, ..., M) - \ell_{t+1}(M_{t-H}, ..., M_t) \right] + \mathcal{O}(\log T)$$

$$\leq \tilde{\mathcal{O}}(poly(\mathcal{L})\sqrt{T}).$$

We provide justification for each step. The first line is the regret definition. Next, we utilize the truncated state history representation of the rewards to upper bound the regret by an error term (from Remark 9) in addition to the approximate costs as a function of $\hat{\mathbf{x}}_t^{\mathcal{A}}$.

Now, because the approximate states are linear in the $\Psi_{t,j}$ and the rewards are quadratic, it can be shown that the rewards for the approximate states are quadratic in the actions. Therefore, even using the truncated history of length $H$, the optimization is still of a form amenable to Lemma 6 and Theorem 7. As such, every step of Alg. 1 becomes efficiently computable.

Finally, efficient computability allows us to use an effective oracle within the Non-convex FPL framework of Suggala & Netrapalli (2020). Therefore, after rearranging to the third line and upper bounding the $1/T$ terms, we arrive at the final line by citing the FPL with Memory result in Ghai et al. (2021). □

## 6 EXPERIMENTS

We demonstrate the effectiveness of our method under different online scenarios and various modes of environmental perturbation. For the experimental setting, we consider a 2D racing problem with the goal of avoiding obstacles in an online fashion (Fig. 1). We use this example to demonstrate cases in which our algorithm outperforms established baselines.

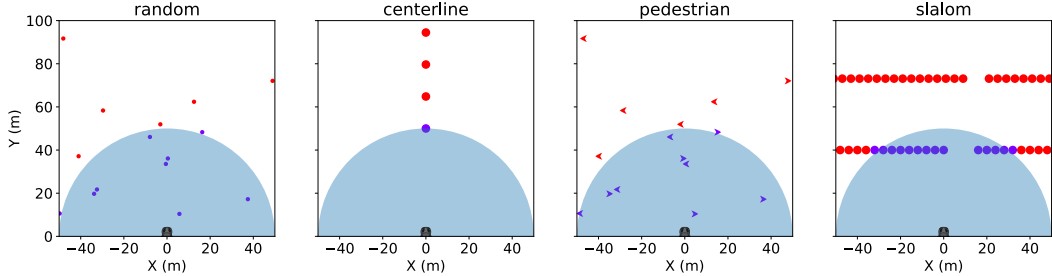

Figure 1: Visualization of experimental environments. (a) Random static obstacles. (b) Vertically stacked obstacles. (c) Moving pedestrian. (d) Slalom course.

### 6.1 EXPERIMENT OVERVIEW, BASELINE PLANNERS, AND SETUP

In the experimental environment, the racer (car) starts at the origin and observes obstacles in its sensor range (blue shaded semi-circle). The obstacles are generated at random positions for every testing environment and can have different sizes and velocities; this adds partial observability to the environment. We implemented our algorithm and the simulation environments in JAX; all experiments are carried out on a single CPU in minutes. On average, 5.6 seconds are required for the computation of a control signal every time a new observation is made.

The experiments consist of two components. First, our planner is evaluated across four different scenarios (see Fig. 1): (1) random static obstacles, (2) centerline static obstacles, (3) "pedestrian" obstacles that have unknown, non-zero velocities, and (4) a challenging "slalom course" with walls of obstacles and gates. For each environment, both stochastic and non-stochastic disturbances are tested and the safety evaluated via measuring the minimal distance to obstacles. The second component of the experiments focuses on the second configuration ("centerline") described above, and explores how our algorithm performs against several baselines. We utilize a HJ reachability planner (ASL, 2021), as well as a modified kinodynamic RRT* implementation (Zhou, 2020) to generate more "optimistic" plans. We hypothesize two key behaviors: our algorithm (1) performs (nearly) as effectively with respect to standard LQR costs in states and inputs as RRT* when disturbances are "benign," and (2) performs (nearly) as well as HJ methods when disturbances are adversarial.

### 6.2 EXPERIMENTAL RESULTS

The first component of the experiments is summarized in Fig. 2, where minimal distance to obstacles over time is plotted for each environment and disturbance type. The distances for all cases remain greater than zero, indicating that our algorithm remains safe and avoids collisions. We note that for challenging slalom instances – those with narrow openings and large requisite deviations – our algorithm suffers empirically due to the inherently long cost memory within the slalom environment design. This corresponds to the planner being forced to overcome a very poor nominal planned trajectory, and in combination with the gated passageways, requires very precise sequences of inputs. This notion of "memory" provides an interesting and interpretable metric of the difficulty of an obstacle avoidance instance, and is discussed further in App. C.3.

For the second component of the experiments, we report the results for each planning algorithm on the centerline example in Table 1. For each planner, three separate disturbance setups were utilized, and each planner-disturbance configuration was run on the equivalent of 50 centerline obstacles. Several key aspects of Table 1 deserve note. First, as expected, RRT* fails to handle unmodeled disturbances

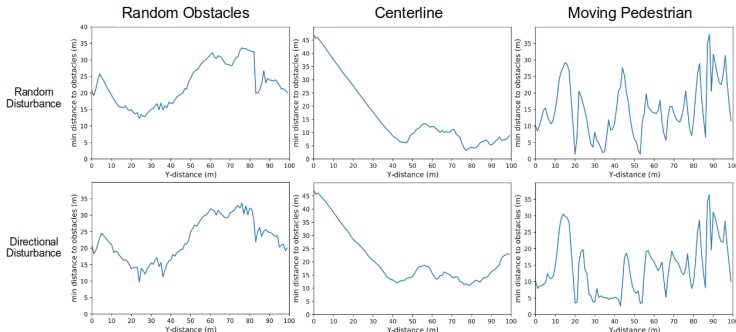

Figure 2: Performance across varied environments and disturbances. The minimal distance remains positive, demonstrating the safety of our planner.

effectively, with high failure rate (see App. C.1). Second, while HJ planning methods improve as adversariality is reduced, they fail to take advantage of existing structure in the disturbances; this is readily apparent in the fact that for sinusoidal noise (which incentivizes the racer to pass the obstacles on the right), the HJ planner passes on the left and right in equal proportion. In contrast, our planning module is able to adapt to the structure of the disturbances to take the "easier route" with lower cost. Sample trajectories for each case are included in App. C.2. Additionally, we note that, while our planner does not recover the performance of RRT* under random disturbances, if the fraction of failures is controlled for (i.e., if we average only over the best 40% of online runs), the performance of our algorithm is $0.42 \pm 0.03$, versus $0.50 \pm 0.03$ for the equivalent for the HJ planner.

Table 1: Planner performance for each disturbance type. Costs are given in terms of linear-quadratic (LQ) costs (top) and fraction of failures (bottom). Best-performing cases for each column are **bold**. LQ costs are only computed for successful passes; as such, RRT* is intentionally blank for two entries.

| **Planner** | Random | Sinusoid | Adversarial |
|---|---|---|---|
| RRT* | $\mathbf{0.27} \pm 0.05$ | —- | —- |
|  | 0.60 | 1.00 | 1.00 |
| HJ Plan | $0.55 \pm 0.05$ | $0.59 \pm 0.14$ | $\mathbf{1.01} \pm 0.03$ |
|  | 0.00 | 0.00 | 0.00 |
| Online (ours) | $0.51 \pm 0.09$ | $\mathbf{0.49} \pm 0.13$ | $1.57 \pm 0.65$ |
|  | 0.06 | 0.04 | 0.26 |

## 7 CONCLUSION AND LIMITATIONS

We develop a regret minimization framework for the problem of obstacle avoidance. In contrast to prior approaches that either assume worst-case realization of uncertainty or a given stochastic model, we utilize techniques from online learning in order to *adaptively* react to disturbances and obstacles. To this end, we prove regret bounds that demonstrate that our obstacle avoidance policy is comparable to the best policy *in hindsight* from a given class of closed-loop policies. Simulation experiments demonstrate that our approach compares favorably with baselines in terms of computational efficiency and performance on settings with varying disturbances and obstacle behaviors. Additionally, our framework provides a possible means of understanding the instance-specific "difficulty" of obstacle avoidance problems through parameterizations of the cost memory ($H$) and other parameters. As such, future guarantees may be possible for classes of dynamics *and* classes of obstacles.

In future work, we intend to explore the above directions in addition to richer comparator classes. Further, it would be of practical interest to tackle settings with more complex dynamics and obstacle geometries than considered here. Finally, we are interested in practical demonstrations of our approach on real hardware platforms (e.g., drones).

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
