# OpenReview forum: "Online Learning for Obstacle Avoidance"
_ICLR.cc/2023/Conference — Submitted to ICLR 2023_

### Official Review · Reviewer_txHg · 2022-10-23

**Confidence:** 2
**Correctness:** 3
**Technical Novelty And Significance:** 2
**Empirical Novelty And Significance:** 2
**Recommendation:** 3

**Clarity, Quality, Novelty And Reproducibility:**

The paper is mostly well-written. Some details mentioned above could be improved. Unfortunately, its main result is a proof in a six page appendix, which I did not carefully check for correctness.

**Strength And Weaknesses:**

Strengths:
- Obstacle avoidance is an important area of robotics, and the proposed approach appears to be novel, based on some recent advances in online learning theory for linear systems
- The paper is mostly well-written
- The paper makes a good effort of citing related works on obstacle avoidance from control and robotics

Weaknesses
- Strong assumptions: From an obstacle avoidance perspective, the significance of the proposed approach is hampered by the assumption that the world is adequately modeled by unconstrained linear dynamics. Most robots have non-linear dynamics (e.g. cars), and especially control saturation constraints (e.g. they cannot generate infinite accelerations on demand). While the proposed approach provides provable bounds, it is not obvious how applicable these would be in practice.
- Weak experiments/baselines: I like that the authors make an effort to compare against baselines from the control/robotics literature. However, worst-case approaches are pessimistic against highly dynamic obstacles like people, and open loop ones that ignore motion and uncertainty are of course very optimistic. Further, the performance of RRT* depends heavily on the number of samples, which is not shown to be sufficient for these experiments. A bigger issue is that the experiment environments appear very simple, and important details are unclear. i) The results across all environment types appear to be averaged together, but these are vastly different problems. To understand what is going on it would be helpful to report them separately. ii) On the experiment whit static obstacles, I do not understand why the RRT baseline would ever crash. iii) On the experiments with moving obstacles, is the random velocity assigned to them fixed at the start of the episode or does it change? iv) Are the obstacle velocities ever estimated and used as inputs of the baseline avoidance algorithms? Work on avoidance of moving obstacles in robotics typically includes estimated motion predictions, and focuses on finding real-time solutions (c.f. [1] and citing papers).
- The abstract claims this is an efficient approach, but in what way? It still collides 4-26% of the time, and it takes 5.6 seconds per time step to compute, which for obstacle avoidance systems seems potentially problematic. I suspect this again refers to the regret bound(?), but they do not really do a good job of showing how this is effective in the experiments. From an obstacle avoidance perspective, it is very unclear what benefit the proposed approach would provide.
- The main contribution of the paper is the provable regret bound, but the proof is deferred to a six page section of the appendix. I understand extra space is sometimes needed for proofs, but this is the main result of the paper.

[1] Wilkie, David, Jur Van Den Berg, and Dinesh Manocha. "Generalized velocity obstacles." 2009 IEEE/RSJ International Conference on Intelligent Robots and Systems. IEEE, 2009.

**Summary Of The Paper:**

The authors propose a novel method for obstacle avoidance by using online learning theory and provide regret bounds on this. The proposed method is claimed to be efficient, provides instance-optimalty to perturbations and compares favorably to baseline open-loop planning as well as a robust planning approach.


**Summary Of The Review:**

This paper builds on recent work in online learning theory for control of linear systems, further extending it to the application of obstacle avoidance. The result seems rather narrow and technical as the assumptions of unconstrained linear systems do not seem like a great fit for obstacle avoidance. Even though the paper does compare against baseline planners from control/robotics, they do not appear to be strong ones (at least in the case of moving obstacles), and the experiments in particular are not convincing from an obstacle avoidance perspective.

That said, the paper is mostly well-written. Unfortunately, the main contribution, the provable regret bounds, is a six page proof unfortunately only found in an appendix. There is clearly a lot of work put into this, but it is difficult to evaluate its significance since it rests on assumptions that usually do not hold in the targeted application. From an obstacle avoidance perspective, it is unclear when this would actually be of use. I am willing to change my mind if somebody willing to champion the theoretical contributions on the online learning side.

---

> ### Author Response · Authors · 2022-11-16
> **(1/3) Re: Assumptions and Experiments**
>
> Dynamics: As a small note regarding dynamics assumptions, it is the case that our results automatically extend to any
> feedback linearizable system (through the linearization in base planning level). However, it still is not as
> general as we might wish, particularly (as the reviewer points out) for non-holonomic systems like many car models.
>
> Experiments: From a conceptual standpoint (before addressing specific concerns below), as noted in our paper, the
> pessimism/optimism problem is something we are trying to address via the “open-loop planner plus
> regret controller" setup. The intuition is that the open-loop planner prevents halting pessimism while
> the regret controller adds robustness. This is why the baselines were selected as HJ methods and RRT* respectively.
>
> With regard to the RRT* sample size - it’s not clear exactly what the intended meaning is here; the density of the
> graph when selecting a path was quite high, and the poor performance is due to the lack of robustness
> to disturbances (which are realized with replanning allowed). In terms of resampling, the sampling
> rate for observing disturbances was set to be equal to all of the other methods. However, again, we are not
> sure as to the specific intent of the reviewer's statement, and would definitely discuss this further if desired.

---

> > ### Comment · Reviewer_txHg · 2022-11-28
> > **Thanks for the additional information**
> >
> > Some details are more clear to me now, but they also confirmed that the paper does not adequately demonstrate how the contributions are actually useful for obstacle avoidance. As the other reviews were also mixed on of the significance of the online learning theory proof, I cannot raise my score.
> >
> > As the authors note, this paper has an identity problem. If this should be seen as a useful complement to an open-loop planner in actual obstacle avoidance, it would help if it had more realistic experiments and baselines showing this. A very simple robot that crashes into static obstacles due to some noise and no safety margin, or having moving obstacles that bounce on walls with constant velocity, is not convincing from an obstacle avoidance perspective.
> >
> > Regarding the provable aspects, this can be valuable but it is not obvious to me how minimum regret transfers to safety metrics for obstacle avoidance, where one is typically interested in guarantees on safety (i.e. not crashing). If the paper wants to improve the obstacle avoidance side, I suggest the authors give some thought on how to demonstrate its usefulness in this application.

---

> ### Author Response · Authors · 2022-11-16
> **(2/3) Re: List of comments [i), ii), ... v)]**
>
> i): The results in the table are specifically for the centerline example. This was noted in the text, but not
> in the table caption. This will be fixed in our revisions.
>
> ii): Crashing is due to disturbances in the dynamics causing deviations from the nominal trajectories.
> The RRT* algorithm is allowed to re-plan upon observing disturbances (in line with the LTI dynamics
> steps, and with each of the other algorithms), but despite replanning it doesn’t utilize enough of a
> margin of safety to avoid crashing (this is probably unsurprising given that margins of safety are not
> explicitly accounted for in RRT*)(we note in the Appendix that we didn't use, e.g., obstacle padding as a heuristic safety
> margin).
>
> iii): In this example, the speed is fixed but the obstacles reflect off the boundary (so they go back and
> forth). The intention of this experiment was primarily to show that (1) obstacle motion could be
> handled well, and (2) that many obstacles could be incorporated into the optimization. It was not the
> case that this was designed to reflect otherwise ‘human-like’ or ‘intentional’ behavior.
>
> iv): We do not generally assume access to obstacle velocities. This was semi-intentional in an effort to
> minimize necessary information. We readily concede that RVCA (and ORCA / Voronoi tessellation
> and related algorithms) outperform ours; our intention was not to be state-of-the-art but to provide a
> means of tractable, guaranteed reasoning about such instances. Also, the omission of some of these methods from
> the related work was an editing oversight that we will fix in an updated version. We want to stress,
> though, that our method is designed to allow provable statements about the obstacle avoidance problem
> instance and control algorithm, and that it is not designed at present to achieve state-of-the-art performance.
>
> v): The efficiency is meant in the sense of formulating the instance as a convex optimization problem,
> which allows the obstacle avoidance synthesis to be done online with low regret. In response to
> reviewer vUNR, we noted that in terms of practical implementation we did not use any warmstarting,
> parallelization / GPU implementation, nor efficient obstacle paring-down, all of which should
> significantly speed up convergence due to the physical (continuous) nature of the problem.
>
> In regards to the utility of such a framework, we intend to discuss more fully its implications,
> especially in contrast with existing guarantees (see the discussion of probabilistic guarantees given to
> reviewer vUNR, above).

---

> ### Author Response · Authors · 2022-11-16
> **(3/3) Re: Location of the Regret Bound Analysis**
>
> This is related to a repeated theme in some of the reviews. In writing the paper, we envisioned the
> contribution to be, in substantial part, a proof of a regret bound for a reasonable (if not the exactly
> standard) obstacle avoidance setting with interesting possible formulations of safety statements (see
> response to reviewers YMGQ and vUNR, above). However, the technical novelty needed to achieve
> this was primarily related to reducing the problem to an existing online learning framework ((Ghai
> 2021) building on (Agarwal 2019)) with further synthesis of additional results in online learning
> and games (Hazan 2006). However, given this reduction, the mechanics of the regret proof are less
> important because they mostly follow (Ghai 2021, Agarwal 2019, Hazan 2006) in outline. As such, some of
> the other reviewers perhaps wished to see less of the proof... In summary, while we would like to
> to highlight the specific technical challenges we address, along with the synthesis method,
> we may actually move the remaining details of the proof out of the paper. Nonetheless, we note that, in line
> with this review, the balance and emphasis of results in our paper “struggles a bit for identity” at the
> moment, and we acknowledge the reviewer's feedback on this point.

---

### Official Review · Reviewer_GFpG · 2022-10-27

**Confidence:** 3
**Correctness:** 3
**Technical Novelty And Significance:** 2
**Empirical Novelty And Significance:** 2
**Recommendation:** 6

**Clarity, Quality, Novelty And Reproducibility:**

This paper is well-written and easy to follow in general, while I suggest the authors to expand Section 3.3 about the trust region optimization. Specifically, when can the optimization problem (6) be solved efficiently? Currently, the readers need to check the references to understand this part. As I discussed in my answer to the previous question, I think the main novelties of this work are the problem setting and the iterative subroutine to solve optimization problem (8).

**Strength And Weaknesses:**

Strength:

Online control has received much attention recently and a limitation of previous works is that the sublinear regret guarantees usually relies on the convexity of the cost functions. It is good see the sublinear regret bound still holds for the obstacle avoidance problem, where the cost function is nonconvex (though it has a special form).

While the FPL type of online control algorithm has been used in, e.g., Ghai et al. (2021), a novelty in Algorithm 1 is the iterative subroutine to compute (8). Unlike the settings in previous works, since the authors consider the minimum distance to any one of the obstacles, the optimization problem (8) is not a trust region problem, thus one cannot solve it directly.

Weakness:

My first concern is about the controller class (or parameterization): It is unclear to me that the disturbance-action controller class is expressive enough for the task of obstacle avoidance. In my opinion, an alternative controller class like Model Predictive Control (MPC) provides more flexibility to avoid a set of moving obstacles, whose positions are not included in the historical disturbances that a disturbance-action controller can react on. Thus, my questions are: Is it possible to extend the analysis in this work to other controller classes like MPC? What is the main technical difficulty of doing the extension?

I also have a concern about the necessity of introducing this specific setting of obstacle avoidance. Specifically, the setting in Agarwal et al. (2019) can already handle general convex stage cost functions. In contrast, if we incorporate the reward for avoiding obstacles into the original stage costs, the new stage costs still have a specific “quadratic” form though it is nonconvex. Therefore, my questions are: If we directly apply the online control algorithm in Agarwal et al. (2019) to nonconvex stage costs, do we expect a counterexample to show the regret will be much worse? If we solve the nonconvexity problem by some convex relaxation approach to the nonconvex stage costs, will the performance loss be significant?

Lastly, I have a slight concern about the way that the problem setting penalizes a direct “collision” with the obstacle. In many applications, I believe it is more natural to consider the obstacles as hard constraints and model them as a compact set rather than singletons. I think that might bring much technical difficulty to the proof, but I recommend the authors to add more discussion about the specific challenges, because I believe online control with hard constraints on states and inputs is an important open question.

**Summary Of The Paper:**

This paper studies an online control setting where the online controller is encouraged to stay away from the obstacles. The authors consider a linear time-invariant system with time-invariant quadratic costs and the class of disturbance-action controllers. Despite the control costs that penalizes the state and control inputs in classic online control settings, the online controller also receives a reward based on its distance to the nearest obstacle. To compete against the clairvoyant optimal disturbance-action controller, the authors proposed a Follow-the-Perturbed-Leader type algorithm with an iterative subroutine that solves a sequence of trust region objectives. The main result is that the proposed algorithm achieves a sublinear regret bound against the clairvoyant optimal disturbance-action controller. The authors also used numerical simulations to demonstrate the effectiveness of the proposed algorithm.

**Summary Of The Review:**

In summary, I think the setting and the algorithm in this paper will be interesting for the online control community. But my concern is that some parts of the problem setting (time-invariant LQR costs/dynamics and disturbance-action controller class) are relatively simple and many previous works on online control have already considered more general costs/dynamics/controllers, so the main results are not significant enough. Therefore, I would like to vote for a borderline accept.

---

> ### Author Response · Authors · 2022-11-16
> **(1/3) Re: Controller Parameterization**
>
> This is definitely an extension we have in mind, and in some ways the major end goal of this
> framework. The primary challenge is the proper parameterization of the obstacle positions in a way
> that is ‘practically logical and useful while remaining convex.’ Namely, the simplest, naive option is
> to parameterize obstacle positions linearly (then the mathematical extension is direct and immediate).
> However, this results in the undesirable effect of having farther-away obstacles create larger control
> inputs. Thus, that option in particular is poorly interpretable and not practical.
>
> We think that with some analogies to a potential-function like approach (a la barrier functions,
> potential-based control, etc.), we could create an ‘inverse-relative-position space’ parameterization
> from which we could reason about collisions in said space while applying linear feedback and
> attaining interpretable guarantees. The main challenges are - as always - (1) interpretability / utility
> of an arising guarantee, and (2) ensuring that the optimization hasn’t become too much more complex
> as a result (i.e., ``nonconvexifiable"). Intuitively, the challenge lies in the feedback component - the effect of the obstacle feature on the ego
> agent dynamics through control inputs, and then back again on the resulting obstacle features.

---

> ### Author Response · Authors · 2022-11-16
> **(2/3) Re: Using Agarwal 2019 Method**
>
> We want to rephrase the question slightly, to "how does this compare to (Ghai 2021)?" -- from which we will
> return to Agarwal.
>
> To be clear: we are not trying to claim that we have extended the theory of online learning beyond
> that of (Ghai 2021). Instead, we have worked to reduce an obstacle avoidance problem to one that can
> be solved with low regret via the FPL with Memory approach. The key technical points for doing this
> synthesize several existing results in online learning (and require some additional math on their own)
> in order to do the reduction. Concretely, the avoidance problem includes ‘switching constraints’ (see
> responses to Reviewers YMGQ) that match any possible action to an encoding
> of the resulting nearest obstacle(s) for that action. The switching causes nonconvexity that is not
> trivial (in our view) to solve. In order to circumvent this, we show that a softened-encoding relaxation
> is expressable itself as a ‘vanilla’ trust region problem, and that (using the results in (Hazan 2006)
> along with regret bounds for exponentiated gradient) the sequence of trust region instances created
> via iterative optimization converges to the true, integer obstacle encoding. This trust region instance, which
> is the limit of a sequence of trust region instances, is what is then implementable in the method of
> (Ghai 2021).
>
> The intuition for how we differ from (Ghai 2021) is the work to specify the input for the trust region solver, which is
> not direct and indeed requires a limit instance of a relaxed trust region problem. Then, to the extent that (Agarwal 2019)
> works for the setting of (Ghai 2021), it works for us as well. For some intuition into why it might or might not: (Ghai 2021) uses
> trust region (2nd-order) methods for making guarantees because it is a kind of "anti-control" problem (drive a system away from a fixed point). (Agarwal 2019), on the other hand, uses gradient (1rst-order) methods, because it is, conversely a "control" problem (driving a system with convex costs to an fixed point). In practice, the gradient methods could work quite well, but they can't readily be used in the analysis to make guarantees.

---

> ### Author Response · Authors · 2022-11-16
> **(3/3) Re: Fully Hard Collisions**
>
> This is a good point, and we agree with the value of adding more discussion of the fundamental
> frameworks of obstacle avoidance and the technical implications thereof. In short, we are not sure of
> an effective means of achieving a (meaningful) regret bound with hard (nonconvex cost) collision constraints.
> The goal of our framework is not to supplant methods that deal with hard constraints, but to
> add complementary perspectives and reasoning capacity. For example, in the potential extension
> referenced in the first part, we envision a regret guarantee in which a subset of obstacles or a subset
> of fields of obstacles - those with low enough ‘memory’ (e.g. ones that are sufficiently small, slow,
> sparse, etc) are guaranteed to be safe, which allows for a certain kind of difficulty that intuitively is
> present in considering the obstacle avoidance task.
> Such regret-like guarantees could for example be used as probabilistic statements as well, under
> perhaps the added distributional assumption that (large, fast, dense, pathological) obstacles / obstacle
> fields are sufficiently rare.

---

### Official Review · Reviewer_Mcwo · 2022-10-27

**Confidence:** 2
**Correctness:** 3
**Technical Novelty And Significance:** 3
**Empirical Novelty And Significance:** 3
**Recommendation:** 6

**Clarity, Quality, Novelty And Reproducibility:**

The paper is fairly clear, but it is difficult to follow the more technical parts. Algorithm 2 has inputs that are not easily mapped to Eqn. 8 (vectors a_j and b for example)..

The quality and novelity seems high. Then again, I would not use neither RRT* (too slow) or HJ planning for dynamic obstacle avoidance (nominal trajectory tracking or otherwise), but rather learning-based MPC [1].

The reproducibility does not seem easy to me, but I am not as experienced in implementation of some of the technical aspects of the paper. Releasing the code upon accept is highly encouraged.

[1]  L. Hewing, K. P. Wabersich, M. Menner, and M. N. Zeilinger, “Learning‐based model predictive control: Toward safe learning in control,” Annual Review of Control, Robotics, and Autonomous Systems, vol. 3, pp. 269–296, 2020.

**Strength And Weaknesses:**

Strength:
* The approach is possible to analyze formally and certain guarantees as well as performance measures (i.e. regret bounds) are proven.
* The approach is straight forward to integrate into motion planning achitectures and interface with a planner that produce a nominal trajectory. Even if the nominal trajectory is not collision free at plan time, then the proposed method will perform collision avoidance.


Weaknesses:
* The approach seem to be too slow for (soft) real-time performance with current hardware and potentially vary a lot in computation time? (5.6 seconds on average). Since collision avoidance is safety-critical in most applications, then an upper bound on time is usually enforced (after which it does not matter any more anyways). How does the approach deal with an upper bound on time (even if it is mostly finished before the bound is reached)? How will the approach perform if the horizon is reduced to reach 1hz (or stay below the upper bound with high probability)?
* The problem formulation in (2) where the distance to obstacles is supposed to be minimized (with relative importance with nominal tracker error and control signal penalties) seem a bit simplistic and hard to tune. If it is desired that the probability of collision is less than say 10^-5 each second, then the only way is to achieve this is to reduce the penalties on the other terms which would then hold globally. However, the collision probability is time (and state)-dependent: It is less likely the further away, and it is less likely if the disturbance is less adverserial etc.. How would such safety constraints or considerations fit into your approach and/or extensions thereof?

**Summary Of The Paper:**

The authors propose a method where a control policy, to simultaniously follow a nominal trajectory and to keep obstacles at a distance, is learned online as to adapt to the disturbance characteristic that is present for the given problem instance.
Algorithms for doing so are introduced and regret bounds are proven.
The approach is evaluated on four toy-environments with varying difficulties and chracteristic of the disturbance. The proposed method is shown to be competitive with a kinodynamic RRT* and a Hamilton-Jacobi reachability planner.

**Summary Of The Review:**

The paper propose an interesting way to deal with obstacle avoidance for nominal trajectory tracking when much is unkown about the disturbances. Relevant proofs are provided and the approach is evaluated with competing methods while showing benefits in some regards over these.

---

> ### Author Response · Authors · 2022-11-16
> **(1/2) Re: Speed of the Approach**
>
> This is a very good question. Our primary focus for the purpose of this paper was to establish the low
> regret (via demonstrating convexity) of the problem being addressed. Given that the problem can be
> specified efficiently, this should be sufficient to ensure efficient computability (i.e. polynomial-time
> scaling). Our interest was not so focused on ‘practical’ speed.
>
> With that said, while the computation at present is too slow to be used in real time, we did not (per
> the preceding paragraph) devote much time to accelerating practical implementation. We ran the
> optimization naively and unparallelized on a single (albeit relatively fast) CPU with Jax. We believe
> that there is significant potential for speedup in the following areas:
>
> - Warm-starting
> - Heuristic Obstacle paring
> - Parallelization / hardware improvement
>
> As a brief summary: warm-starting from previous solutions should improve convergence because the dynamics are continuous and thus nearby obstacles are likely to remain nearby for the short term. Obstacle paring (get rid of far away obstacles if nearer ones are clearly binding) reduces the problem instance sizes, potentially significantly. And of course, better hardware is better hardware (given that matrix computations are involved, we think this offers non-trivial potential as well).

---

> ### Author Response · Authors · 2022-11-16
> **(2/2) Re: Weakness 2 - Probabilistic Guarantees and More Explicit Forms of Safety**
>
> This is a fundamental question for the obstacle avoidance problem, and we agree that we should
>  address it in more detail. There are a few ways in which we think (1) a regret measure
> provides an additional, complementary perspective on reasoning about such algorithms, and (2) provides alternative
> statements of safety that might be of practical use.
>
> The specific case listed here - making a statement like "the probability of collision at every [decision
> step] is at most 10e-5" - reflects a fundamental point about obstacle avoidance (and safety in robotics
> more generally) that we think is very important. Namely: what kind of statement is (1) meaningful
> / useful, and (2) feasible / computable? For probabilistic statements, some common challenges
> include (a) dealing with rare events, and more fundamentally (b) making either an explicit model
> of the uncertainty [in order to explicitly control / reason about the moments of the probability distribution], or assuming
> possibly-strong statements [like i.i.d. instances at each decision step]. Additionally, evaluation at very
> low probabilities can be data intensive, in part due to (a), above. It is for this reason that
> we feel a regret framework can be a useful complement to the existing methodologies (while
> conceding that downsides of regret techniques very much exist as well, and this paragraph could just as easily
> be directed at our method).
>
> To address the direct question, it is unlikely that probabilistic statements in particular would integrate
> cleanly with our method, in part because the guarantees are designed to be nonstochastic. It is
> possible that one could attempt to evaluate our method in a setting where such statistical methods are
> valid, but that wouldn’t be our method itself. Rather, our method would (ideally) make deterministic
> statements but with respect to a weaker class. E.g., our basic guarantee at present is “we guarantee
> that we will perform with sublinear regret compared to the best parameterized controller in hindsight.”
> A more interpretable version of this, if we can extend to the case in Reviewer GFpG’s first comment
> (which we hope to do - see below), would be (ideally) "we guarantee that we will learn to avoid all
> obstacles falling into a certain range of (sizes, speeds, etc)." This is the sense in which we provide an
> alternative (and possibly preferable in some cases) regret-type guarantee for obstacle avoidance - one that
> includes (via regret) an inherent notion of difficulty of the instances observed.

---

### Official Review · Reviewer_vUNR · 2022-11-02

**Confidence:** 2
**Correctness:** 3
**Technical Novelty And Significance:** 3
**Empirical Novelty And Significance:** 3
**Recommendation:** 3

**Clarity, Quality, Novelty And Reproducibility:**

The author is well-organized with a clear structure. The related work is well-written, especially for readers (like me), who are not very familiar with the field of robust planning. One issue is that the experimental analysis is not enough to support the low-regret claim. And it seems that the major contribution (the non-convex memory FPL algorithm) mainly comes from previous work. The reviewer believes that the results of the experiment are reproducible.

**Strength And Weaknesses:**

Strength:
1. The author does not make any assumption on the distribution of uncertainty and proposes a more generic pipeline using regret optimization.
2. The regret objective is well-formulated and an iterative algorithm based on Hidden-Convex is developed to solve it.
3. Regret bound of the problem is given with theoretical proof.

Weakness:
1.It seems that the method part is very similar to the related work cited in the paper: Generating Adversarial Disturbances for Controller Verification. Could the author provide more clarification on this?
2.Experimental comparison to RRT* seems not good: Even though the RRT* baseline is an oracle without partial observability, the visible region is still very large, which covers more than half of the obstacles. In this case, the naive RRT* (as mentioned in supp C1) can still outperform the proposed method by a large margin on the first task.

**Summary Of The Paper:**

In this paper, the author formulated obstacle avoidance as a regret-minimization minimization problem and proposed a gradient-based online learning algorithm to solve it. The author theoretically shows the instance-optimality to perturbations. Compared with previous work, this paper does not rely on the a priori assumptions about whether the uncertainty is stochastic or adversarial and thus allows for more flexible behavior and adaptations. Finally, the authors demonstrate the experimental comparison with an open-loop planner, RRT*, and a worst-case planner, Hamilton-Jacobi (HJ) planner.

**Summary Of The Review:**

First, the reviewer has not have much experience reviewing papers in the field of control theory. Thus some of the comments are subjected to small mistakes. Please correct me during the rebuttal phase if any of my statement (e.g., weakness) is not well-grounded. Thus, the final score of this paper will also be largely modified based on the comments and discussion from other reviewers after the rebuttal phase.

---

> ### Author Response · Authors · 2022-11-16
> **(1/2) Re: Methodology in Relation to Prior Work**
>
> The online learning method utilized here is similar (from the online learning perspective) to the
> cited paper. The key difference, in our view, relates to the application and associated challenges of a
> second-order method in this obstacle avoidance regime. Namely, we now have a trust-region problem
> in which the relevant constraints exhibit a discrete switching behavior (between the encoding of the
> relevant nearest obstacles) as a function of the action selection.
>
> For more detail on the contributions therein, please see the last paragraph to comment (1/2) for reviewer YMGQ.

---

> ### Author Response · Authors · 2022-11-16
> **(2/2) Re: Experimental Comparison to RRT***
>
> We also would have liked to observe our method obtaining a better cost (at the expense of safety) in
> order to illustrate a true ‘recovery’ of RRT performance, and to illustrate the tuneable nature of our
> framework.
>
> We believe that the reason that this was not recovered in practice was due to some fundamental
> pessimism / conservatism baked into the instances of the problem that we ran. Namely, in every
> instance for the costs we report, the RRT* algorithm - in addition to having full map information -
> is optimizing the ‘true’ cost (which uses a hard 0-1 encoding for collisions), whereas our method
> is using a quadratic cost, which incentivizes a larger margin around obstacles. As such, we incur
> larger state and input costs due to deviating farther from the nominal trajectory. We are interested in running experiments that increase the input and
> state deviation penalty costs, and (in the limit as they dominate the collision penalty) verify that our
> algorithm converges more closely to RRT*.

---

> ### Comment · Reviewer_vUNR · 2022-11-29
> **Reply to Author Response after Rebuttal**
>
> Thanks for the explanation and response to the review. I appreciate the effort spent by the author during rebuttal phase. However, I am sorry that I have to lower down my rating after rebuttal. I think I have overestimated the novelty and contribution in the original review. After reading the comments from other reviewer and their replies, I think there is still a large space for this paper to be improved.
>
> In general, I think this paper has little novelty compared with other paper on the theory side, and also not show sufficient evidence to be considered as a good application paper. As mentioned by reviewer txHg and YMGQ, the problem formulation is a direct non-convex FPL where the author does not touch any additional challenge for the optimization. The trust-region problem is not a significant difference between this paper and the prior work.
>
> From application perspective (obstacle avoidance), I am not convinced by the *(2/2) Re: Experimental Comparison to RRT\**. The author mentioned that larger deviation error comes from a soft quadratic cost while RRT apply the 0-1 encoding. It makes sense but reflect the intrinsic problem of the approached. Since RRT utilize sampling to search for a trajectory, discrete loss like 0-1 collision flag can be well handle. But the proposed method needs a continuous parameterize. The tuning of the balance weight also increase the burden for collision avoidance application. Direct increasing the input and state deviation penalty costs (as authors mentioned in reply) have a great chance to make the problem intractable.
>
> Another point is that maybe the author can compared with gradient based obstacle avoidance approaches besides sampled based method which is more similar with the proposed method.

---

### Official Review · Reviewer_YMGQ · 2022-11-03

**Confidence:** 4
**Correctness:** 2
**Technical Novelty And Significance:** 2
**Empirical Novelty And Significance:** 1
**Recommendation:** 1

**Clarity, Quality, Novelty And Reproducibility:**

The studied problem of learning-enabled safe control with unmodelled uncertainty is an very interesting topic and has gained significant attention lately by the learning and control community. The paper is in overall well written to highlight the motivation, existing challenges, and the adapted approach of the non-convex memory FPL framework. With that being said, however, the presented framework seems to be a simple contextualization of the existing FPL framework, with incremental changes of the defined objective function to adapt to the collision avoidance problem. The main proofs of the bounded regret is natural unleashed from standard FPL framework, making the contribution too incremental. Please see detailed comments as follows.


- Contribution and Novelty: while the presented motivation is clear that focuses on episodic online learning for safe control to address unmodelled uncertainty, the reviewer feels the contribution is quite incremental, given that the Algorithm 1 mainly follows the standard non-convex memory FPL and it is difficult to see what is the additional challenge given the targeted application of collision avoidance. The main results of the bounded regret also directly comes from the one for FPL from [Ghai et al 2021, Agarwal et al 2019a, Hazan et al 2020].

- Quality: as a follow-up comment from above, the challenge of collision avoidance itself is not well addressed by the contextualized approach of FPL. For example, there has been extensive results on safe reinforcement learning for robotic control that requires high probability or zero violation of safety during both learning/exploration and execution. The interesting tension comes from how to enforce safety such as collision avoidance when the uncertainty is not fully modelled. In this paper, however, safety is simply embedded as a soft constraint in the objective function with all properties inherited from the low regret FPL, which may make it not applicable in the real world applications of safety-critical control.

- Experimental results: from the reported results, it seems the experiments are only performed with one or a small number of trials and thus makes the results questionable to well support the claimed advantages. On the other hand, there is no metrics that reflect the learning efficiency such as regret.


**Strength And Weaknesses:**

Strength:

+ The paper is generally well written with clear motivation and good overview about the high level idea of the algorithm.
+ A good coverage of the related work in multiple relevant fields.
+ The idea of contextualizing standard online non-convex FPL framework in robotic planning is interesting.

Weakness:

- The technical contribution seems rather incremental as the adapted non-convex memory FPL is a standard approach in many existing work.
- For the target application of obstacle avoidance, there is no formal discussion or proofs in terms of the safety assurance.
- As inherited from the existing non-convex memory FPL framework, most of the proofs are from existing work including the performance on the bounded regret, and it’s unclear what are the novel proofs beyond the existing work.
- There is no comparison between the proposed method and other existing learning-based approaches.


**Summary Of The Paper:**

This paper addresses the problem of safety assurance in terms of collision avoidance for a robotic system under uncertainty learned online. The key contribution of this work lies in the introduced online learning algorithm for safe control in the context of regret minimization, followed by the formal proof of the bounded regret and safety guarantee. Compared to other existing online learning for safe control frameworks, the proposed method relies on the setting of regret minimization to achieve instance-optimality, which allows for less conservative behaviors given the observed stochastic/non-stochastic disturbances in hindsight. Experimental results on a racing car simulation example are provided to demonstrate the effectiveness of the proposed algorithm.

**Summary Of The Review:**

While the studied problem of online learning for safe control is interesting, the paper mainly follows the standard approach of non-convex memory FPL with incremental contribution in applying it to the collision avoidance scenario. On the other hand, the challenge of enforcing safety during learning, as studied in many existing work on safe learning, is not addressed in the paper. The results are also seem substandard given the lack of sufficient number of testing trials. Authors are encouraged to address the comments in the future version of the paper.

---

> ### Author Response · Authors · 2022-11-16
> **(1/2) Re: Contribution + Novelty**
>
> In reading reviewer comments, we note that the presentation in our paper perhaps did not clearly
> distill the technical novelty of our approach from the broader method we present. By that, we mean
> that the broader method is a regret bound for an interesting and meaningful approach to providing
> guarantees about obstacle avoidance problems. The technical novelty in our method arises in two key
> areas: (1) reduction of the model of an obstacle avoidance problem to a trust region instantiation,
> which requires (2) additional synthesis of results from (Hazan 2006) to the existing literature on
> online learning for control (Agarwal 2019, Ghai 2021, Hazan 2020, etc).
>
> The reviewer is correct that we do not make general extensions to the theory of online learning itself
> (a la FPL with Memory extensions in (Ghai 2021)), and as such the space lent to details of the regret
> bound likely obscures our contribution.
>
> The conceptual challenge to the obstacle avoidance setting (that is not present in any of the applications
> of the above resources) is the discrete switching constraint imposed by the selection of the nearest
> obstacle given the action (that is, for continuous action spaces, there will be decision-boundary-
> like surfaces where the nearest obstacle constraint will ‘switch’). The optimization thus contains
> integer-like constraints that one-hot or k-hot encode the nearest obstacle(s) for any given action.
> Surmounting this requires the extension of the trust region subroutine in (Ghai 2021) to the limit
> trust region instance arising out of a specific sequence of relaxed trust region instances. The result
> in (Hazan 2006) allows us to ensure that the sequence of relaxed instances converges to the desired
> limit, and direct analysis of the relaxed instances allows us to ensure that they are valid, ‘vanilla’ trust
> region problems suitable to be used with (Hazan 2006). Having ensured these aspects, much of the
> remaining procedure (as we note in the paper and as is noted in the review) follows from (Ghai 2021).

---

> ### Author Response · Authors · 2022-11-16
> **(2/2) Re: Quality + Obstacle Avoidance Setting**
>
> The 'proper setting' for analyzing obstacle avoidance is a fundamental question, and we do not think that many of the
> standard methods (and the results they have achieved) are misdirected or “not useful.” However,
> in the cases that they wish to make guarantees, they tend to suffer from restrictive assumptions of their own.
> For example, some RL techniques require finite action - and possibly finite state - spaces for their
> theoretical results; as this doesn’t generally scale well to higher dimension, they lose theoretical
> guarantees in practice (even though the methods usually perform well). Similarly, statistical methods often have to assume the structure of the
> uncertainty or require a lot of data (and thus, failures) in the learning process. Further, many such
> cases do not also build in resilience to adversarial disturbances, noise, etc. We might of course be
> missing some examples and would be curious if there are specific cases the reviewer has in mind
> for safe RL in particular.
>
> As such, we will try to express the kinds of guarantees a nonstochastic regret-type method might
> allow with respect to a comparator class. As an example: “for obstacles / obstacle fields that are
> sufficiently (small, slow, sparse), low regret implies that we will find a policy that successfully avoids
> collisions." In the case that an extension to obstacle feedback is undertaken, [see response to Reviewer
> vUNR] such a semantic guarantee might be feasible. We feel that this nature of guarantee more
> directly accounts (via regret) for the notion of ‘difficulty’ inherent in any obstacle avoidance problem -- how sparse, big, fast, predictable, etc all the obstacles are, and meaningfully distinguishes the setting
> from the guarantees arising via other domains, and possibly complements their approaches (without, we agree, necessarily superseding them).

---

### Author Response · Authors · 2022-11-16
**Thank You to the Reviewers**

We would like to quickly include a "meta-comment" thanking all of the reviewers for their detailed and engaged feedback. Direct responses have been made to each reviewer accordingly.

---

### Decision · Program_Chairs · 2023-01-20

**Decision:**

Reject

**Justification For Why Not Higher Score:**

After the rebuttal 2 reviewers argue strongly against this paper, 2 reviewers have a weak accept.
The rebuttal was helpful to calcify a lot of points about this paper, however the in this case highlighted more clearly that the proof is not a major contribution. Some of the reviewers still have doubts how limiting the assumptions will be to make it work in a real obstacle avoidance scenario.

**Justification For Why Not Lower Score:**

N/A

**Metareview: Summary, Strengths And Weaknesses:**

summary:
The paper extends FPL to a simplified obstacle avoidance setting. The approach is validated experimentally.

strengths:
- well written paper
- sound idea
- good literature overview

weaknesses:
- the proposed method is an extension of non-convex memory FPL for obstacle avoidance - the authors said in the rebuttal there is essentially no theoretical contribution (besides reducing the novel setting to a known one)
- the reviewers still think the considered assumptions/setting is too simplified for real world obstacle avoidance
- still doubts about RRT* baseline

**Summary Of Ac-Reviewer Meeting:**

N/A